# Biomarkers for Alzheimer’s Disease (AD) and the Application of Precision Medicine

**DOI:** 10.3390/jpm10030138

**Published:** 2020-09-21

**Authors:** Walter J. Lukiw, Andrea Vergallo, Simone Lista, Harald Hampel, Yuhai Zhao

**Affiliations:** 1LSU Neuroscience Center, Louisiana State University Health Science Center, New Orleans, LA 70112, USA; yzhao4@lsuhsc.edu; 2Department of Cell Biology and Anatomy, LSU-HSC, New Orleans, LA 70112, USA; 3Department of Ophthalmology, LSU Neuroscience Center, LSU-HSC, New Orleans, LA 70112, USA; 4Department Neurology, LSU Neuroscience Center, LSU-HSC, New Orleans, LA 70112, USA; 5Sorbonne University, GRC no 21, Alzheimer Precision Medicine (APM), AP-HP, Pitié-Salpêtrière hospital, F-75013 Paris, France; a_vergallo@yahoo.com (A.V.); slista@libero.it (S.L.); harald.hampel@med.uni-muenchen.de (H.H.); 6Brain & Spine Institute (ICM), INSERM U 1127, CNRS UMR 7225, Boulevard de l’Hôpital, F-75013 Paris, France; 7Department of Neurology, Institute of Memory and Alzheimer’s Disease (IM2A), Pitié-Salpêtrière Hospital, AP-HP, F-75013 Paris, France

**Keywords:** Alzheimer’s disease (AD), biomarkers, diagnostics, messenger RNA, microRNA, neuroimaging, neurotropic microbes, precision medicine, prognostics

## Abstract

An accurate diagnosis of Alzheimer’s disease (AD) currently stands as one of the most difficult and challenging in all of clinical neurology. AD is typically diagnosed using an integrated knowledge and assessment of multiple biomarkers and interrelated factors. These include the patient’s age, gender and lifestyle, medical and genetic history (both clinical- and family-derived), cognitive, physical, behavioral and geriatric assessment, laboratory examination of multiple AD patient biofluids, especially within the systemic circulation (blood serum) and cerebrospinal fluid (CSF), multiple neuroimaging-modalities of the brain’s limbic system and/or retina, followed up in many cases by post-mortem neuropathological examination to finally corroborate the diagnosis. More often than not, prospective AD cases are accompanied by other progressive, age-related dementing neuropathologies including, predominantly, a neurovascular and/or cardiovascular component, multiple-infarct dementia (MID), frontotemporal dementia (FTD) and/or strokes or ‘mini-strokes’ often integrated with other age-related neurological and non-neurological disorders including cardiovascular disease and cancer. Especially over the last 40 years, enormous research efforts have been undertaken to discover, characterize, and quantify more effectual and reliable biological markers for AD, especially during the pre-clinical or prodromal stages of AD so that pre-emptive therapeutic treatment strategies may be initiated. While a wealth of genetic, neurobiological, neurochemical, neuropathological, neuroimaging and other diagnostic information obtainable for a single AD patient can be immense: (**i**) it is currently challenging to integrate and formulate a definitive diagnosis for AD from this multifaceted and multidimensional information; and (**ii**) these data are unfortunately not directly comparable with the etiopathological patterns of other AD patients even when carefully matched for age, gender, familial genetics, and drug history. Four decades of AD research have repeatedly indicated that diagnostic profiles for AD are reflective of an extremely heterogeneous neurological disorder. This commentary will illuminate the heterogeneity of biomarkers for AD, comment on emerging investigative approaches and discuss why ***‘precision medicine*’** is emerging as our best paradigm yet for the most accurate and definitive prediction, diagnosis, and prognosis of this insidious and lethal brain disorder.

## 1. Overview

Senile dementia is the progressive, age-related loss of memory and cognition sufficiently severe to irreversibly affect social, behavioral, perceptual, occupational, and functional capabilities. Recent statistics indicate that globally, about ~50 million people live with dementia, now costing an estimated one trillion dollars in annual healthcare. By 2050, the number of people with dementia is projected to increase to ~130 million. In the United States, Alzheimer disease (AD), the leading cause of senile dementia in the elderly, currently affects about ~6 million people age 65 and older; by 2050, the number of people aged 65 and older with AD will grow to a projected ~15 million if no medical breakthroughs occur to prevent, slow, or cure this incapacitating disorder of the human mind [1,2,3]. One hundred and fourteen years since its original description, despite immense research efforts and clinical trials employing multiple strategic therapeutic approaches, there is currently no adequate treatment or cure for this widely expanding socioeconomic and healthcare concern [2,3,4,5,6,7,8]. 

The discovery, characterization, and quantification of biomarkers as measurable substances or cognitive disruptions in the ***‘prospective AD patient***’, whose presence are indicative of disease, are urgently needed so that: (**i**) AD may be more accurately diagnosed, especially at an earlier ‘***prodromal***’ stage and with the goal of preventive and or targeted therapeutic strategies that may be implemented at the earliest signs of AD onset; and (**ii**) more effective and reliable integration of multi-modal biomarkers for AD that can streamline, support, and strengthen the diagnostic and therapeutic decision-making. Remarkably, peer-reviewed publications on biomarkers for AD have yielded almost ~53,000 original research reports and reviews since 1983 crossing the words ‘***Alzheimer’s disease***’ and ‘***biomarkers***’; [2,3,8,9,10]. These include observations on the classical and established AD biomarkers [11,12,13,14], including altered genetics (incorporating genome-wide association studies or GWAS), genetic mutations and gene modifications (including methylation and post-transcriptional modifications), end-stage neurotoxic and pathogenic metabolic products that accumulate in AD brains, such as multiple forms of tau aggregates and amyloid-beta (Aβ) aggregation species and plaques. AD biomarkers also include protein, lipid, proteolipid, inflammatory cytokine, chemokine, carbohydrate, microRNA (miRNA), and messenger RNA (mRNA) abundance, speciation, and complexity, as well as an evolving assortment of neuro-radiological and neuroimaging technologies (Table 1). AD biomarkers are certainly useful in the detection of dementing illness during its progressive course, and their appearance and magnitude correlate with cognitive loss in a dynamic way, allowing clinicians to monitor responses to therapeutic intervention across a background of aging of the AD patient.

As improvements in AD diagnostics are based on advances in both AD biomarker acquisition and the technologies used to gain these data, below we briefly discuss some of the most recent advances contributing in a major way to the more accurate and comprehensive accrual of important AD biomarker data.

## 2. Novel, Emerging, and Advanced Diagnostic Biomarkers for AD

### 2.1. Analysis of Exosomes (EXs), Extracellular Microvesicles (EMVs), and Their Molecular Cargos

Currently, the complex molecular cargos of exosomes and extracellular microvesicles (EXs and EMVs) have emerged as one of the most representative, significant, dependable, and trustworthy of all AD biomarkers. Typically, EX and EMV cargos consist of various mixtures of protein, lipid, proteolipid, cytokine, chemokine, carbohydrate, microRNA (miRNA) and messenger RNA (mRNA), and other constituents including end-stage neurotoxic and pathogenic metabolic products. These in part, consist of tau proteins, amyloid beta (Aβ) peptides, alpha-synuclein, TAR DNA-binding protein 43 (TDP-43) and others. EXs and EMVs (**i**) have been analyzed in the cerebrospinal fluid (CSF), blood serum, and post-mortem tissues of AD patients; (**ii**) are derived from their cells of origin and typically contain hundreds of different signaling molecules, many of which are potentially pathogenic and may be involved in the horizontal spread of neurological disease from one brain region to another; (**iii**) may represent a defined class of plasma membrane-derived nanovesicles released by all cell lineages of the human central nervous system (CNS); and (**iv**) as potential biomarkers, may contribute to an additional element of certainty into the diagnostic assessment of AD [20,21,26,30,31,32]. 

### 2.2. The Evaluation of Neurotropic Microbes in AD as Potential Diagnostic Biomarkers

There is a wealth of data indicating that neurotropic microbes including both DNA and RNA viruses (such as *Herpes simplex 1* or *SARS-CoV-2*) or bacterial Phyla such as *Proteobacteria*, *Verrucomicrobia*, *Fusobacteria*, *Cyanobacteria*, *Actinobacteria*, and *Spirochetes* or microbe-derived viral, fungal, or prokaryotic cellular components or microbial neurotoxins have high affinity for neural cells of the human brain and CNS [24,33,34,35,36,37,38,39,40,41]. Multiple independent laboratories continue to report the detection of viral, bacterial, fungal, protozoal, or other microbially-derived nucleic acid sequences or neurotoxins, such as highly inflammatory bacterial amyloid peptides, lipopolysaccharide (LPS), and many microbe-derived endotoxins within AD affected brain tissues [24,29,34,36,39,40]. Microbial biomarkers and systems biology approaches to understand host–microbiome interactions have been suggested by multiple AD researchers that both: (**i**) predict the risk of developing AD well before the onset of cognitive decline; and (**ii**) stimulate and/or accelerate the development of classical AD neuropathology [24,34,39,41,42,43,44]. 

Whether these viral, bacterial, or other microbial DNA- or RNA-based nucleic acids or associated lipoproteins, liposaccharides, peptidoglycans, bacterial-derived amyloids, and/or neurotoxins originate from the gastrointestinal (GI) tract microbiome, a potential brain microbiome, or some dormant pathological microbiome is currently not well understood [24,35,36,40,43,45]. Since 1978, at least ~4400 peer-reviewed research articles provide convincing evidence that multiple species of microbes, including viruses, bacteria (especially Gram-negative bacteria), and other microorganisms or their secreted components are strongly associated with the onset and/or the development of AD-type change [24,29,33,34,41,42,43,46]. If microbial presence in the brain is involved in the early initiation or propagation of AD, as currently suspected, then specialized RNA-sequencing applications or nucleic acid-containing gene chips, electrochemical biosensors, or panels of microbial-derived 16S ribosomal RNA (rRNA) interrogated with nucleic acid probes derived from AD biofluids might be useful as novel AD biomarkers in the detection of microbial patterns of expression from human brain tissues at any stage or degree of AD neuropathology.

### 2.3. Linking microRNA-messenger RNA (miRNA-mRNA) Signaling Patterns in AD

DNA microfluidic array technologies, quantitative reverse transcription PCR (RT-qPCR), RNA sequencing, LED-Northern, Western immunoassay, and electrochemical biosensors, integrated by advanced bioinformatics tools have uncovered families of up-regulated human brain enriched microRNAs (miRNAs) and their down-regulated messenger RNA (mRNA) targets. These have been found in short post-mortem interval (PMI) sporadic AD brain, in transgenic animal models of AD (TgAD), in brain biopsies, and in biofluids from AD patients. Genome-wide association studies (GWAS), epigenetic evaluations, such as miRNA-mRNA linkage or association mapping for AD-relevant neurological pathways, should provide useful diagnostic approaches since it has recently become apparent that miRNA-mediated mRNA-targeted regulatory mechanisms involve a large number of pathogenic and highly integrated gene expression pathways in the CNS [25,32,45,46,47,48]. To cite one very recent example, the human-brain-resident, nuclear factor kappa B p50/p65 (NF-κB p50/p65)-regulated microRNA-146a (miRNA-146a) is an inducible, 22-nucleotide, single-stranded non-coding RNA (sncRNA) easily detected in CNS neurons and immunological cell types. An inducible miRNA-146a: (**i**) is significantly up-regulated in AD brain tissues, CSF, and blood serum [25,48]; (**ii**) is an important epigenetic modulator of inflammatory signaling and the innate-immune response in several neurological disorders; and (**iii**) is essential in the down-regulation of the innate-immune regulator complement factor H (CFH; [10,20,23,25,40,46,49]).

LPS- and NF-kB (p50/p65)-inducible microRNAs, such as miRNA-146a and miRNA-125b, appear to contribute to neuropathological, neuro-inflammatory, and altered neuro-immunological aspects of both AD and prion disease (PrD; [25,32,40,46,48,49,50]). Interestingly, NF-κB-sensitive up-regulated miRNAs and their down-regulated mRNA targets appear to constitute an integrated NF-κB-miRNA-mRNA signaling network implicated in multiple AD pathophysiological processes [10,40,45,48,50,51]. Hence, potential signaling pathways to the acquisition of the AD phenotype appear to occur in part via an integrated and highly complex system of multiple miRNA-mRNA interactions that define many key pathogenic and pro-inflammatory gene expression pathways. Genetic and epigenetic signaling via miRNA-mRNA networks in the brain may be one of the most useful as potential biomarkers for early AD detection as they can detect subtle failure in multiple AD-relevant brain signaling systems and metabolic pathways [10,25,45,49,51]. 

### 2.4. Recent Advances in Neuro-Radiological and Neuroimaging Technologies

A number of neuro-radiological and neuroimaging technologies are currently being used to view physical atrophy and structural change in specific anatomical regions of the human brain, such as the hippocampus, neocortex (gray matter), white matter, ventricles, and other brain regions for the purpose of acquiring real time data for the diagnosis of AD [19,28,52,53,54,55,56,57,58,59,60,61]. These neuroimaging technologies and structural and functional imaging techniques include computerized tomography (CT; including dual-energy CT), positron emission tomography (PET), scintigraphic neuroimaging (PET-SN), diffuse optical imaging (DOT), structural and functional magnetic resonance imaging (MRI), including ultra-high field MRI (UHF-MRI), magnetoencephalography (MEG), single-photon emission computed tomography (SPECT), cranial ultrasound, and functional ultrasound imaging (fUS) in the search for anatomically-based diagnostic biomarkers for AD with high accuracy and sensitivity [19,28,57,58,59,60,61,62]. Neuroimaging techniques, hardware and software design, and imaging resolution are being constantly improved, updated, and refined [28,60,61,62]. For example, with high signal-to-noise (S/N) ratios, improved contrast and unparalleled spatial resolution, ultra-high field MRI of ≥7 tesla (T) has been highly successful in imaging the neuroanatomy of highly focused brain regions targeted by AD pathophysiology while providing additional information on morphological, quantitative, and subtle metabolic changes associated with early AD-type pathological alterations [57,61]. In vivo biomarkers for AD performed by recently implemented scintigraphic neuroimaging and employing amyloid binding PET agents along with non-scintigraphic biomarkers from blood (plasma/serum) and CSF have provided unique and novel opportunities to investigate the pathogenesis, prodromal changes, and time course of the disease in living individuals across the AD continuum [19,28,61,62,63]. Imaging technologies have indicated that AD changes in brain tissues begin as much as ~25 years prior to the onset of clinical symptomatology [28,61,63,64]. The opportunities afforded by *in vivo* biomarkers of AD, whether by blood (plasma/serum) or CSF examination or imaging technologies, are beginning to transform the strategic design of AD therapeutic trials by shifting the focus to the preclinical stages of the disease and massively integrating both molecular-based and neuroimaging data [60,61,63,64,65,66,67]. 

## 3. AD Biomarkers and Post-Mortem Neuropathological Examination of the AD Brain

Classically, the diagnosis of AD was a clinic-pathological one and there was a considerable error rate in the clinical diagnosis, especially early in the course of the disease. The differential diagnosis for AD by exclusion was confounded by a great many clinically overlapping neurological disorders including, mainly, MID, FTD, prion disease, tumors, subdural hematomas, neurovascular disruption and disease, and others [4,5,6]. Early neurophysiological diagnostic observations of AD included a diffusely slow electroencephalogram (EEG) and reduced cerebral blood flow [4,5,7]. Early PET studies demonstrated that oxygen extraction in the AD brain was relatively normal, thus tentatively excluding ischemia as a potential pathogenic factor [4,5,64]. Morphological pathological changes including the appearance of amyloid-enriched senile plaques (SP) and neurofibrillary tangles (NFT), widely distributed in neocortex but excluded from the basal ganglia, thalamus, and substantia nigra, and a severe loss of large neocortical neurons, were ‘classical’ diagnostic characteristics of the AD patient [4,5,6,7,8].

Usually at the family or care-giver’s request, post-mortem neuropathological examinations of the deceased AD patient’s brain were routinely performed by qualified AD-trained specialists and neuropathologists and brain tissues were often subsequently provided to AD researchers for further molecular-genetic and biomarker research including the examination for the presence of AD-relevant brain lesions. Light microscopy, NFT and SP amyloid dyes (such as Congo red, Thioflavin S, Thioflavin T and methylene yellow), and antibody-based staining (such as 6E10 and 10G4), the evaluation, density, composition, and quantitation of NFT and SP amyloid, and the examination of blood (plasma/serum) or CSF amyloid were additional indicators of immune- or inflammatory-neuropathology in the individual AD patient, which often contributed to the confirmation of AD in the ***‘prospective AD patient’.*** Currently, in many medical schools in the US and Canada, the post-mortem examination of the AD brain still remains the classical exercise to certify and verify the existence of AD. It is generally appreciated that the application of ***‘precision medicine’***, involving massively integrated data sets of multi-faceted AD biomarkers, data-driven analytical methodologies, and the application of systems theory, will challenge and may eventually supersede the need for the classical post-mortem neuropathological examination of the brain in order to verify and confirm the diagnosis of AD [10,11,12,13,14,15,16,22,59,64,65,68].

### 3.1. Challenges in the Validation of AD Biomarkers

There are inherent problems in current approaches to AD biomarker research: (**i**) as most early reports emphasized just one or at most a few AD molecular-genetic biomarkers, biophysical, and neuroimaging modalities without consideration of the other hundreds or thousands of proteins, peptides, carbohydrates, lipids, or DNA and multiple species of RNA that have been previously implicated as being ‘probable’ diagnostic markers for AD; (**ii**) very often the nature of the acquisition of AD biomarkers represented a ‘***snapshot in time***’ of one specific portion of the AD continuum that does not take into consideration the time course of changes observed in AD and/or the progressive nature of the disorder; (**iii**) because easily accessible and non-invasive AD biomarkers are often limited in their diagnostic applicability because of their overlap with other neurological diseases related to AD, such as Down’s syndrome, Parkinson’s and Lewy body disease, prion disease, FTD, hippocampal sclerosis, and MID and stroke; and (**iv**) no single newly generated ***de novo*** biomarker has ever been found to be associated with AD; that is, fluctuations in the abundance of pre-existing AD biomarkers reflect significant and absolute differences in the quantity, speciation, and stoichiometric relationships of AD-related biomolecules, including indicators of pro-inflammatory and immune system dysfunction. Put another way, no ‘specific’ AD biomarker ‘suddenly appears’ at the earliest onset, or propagation, or throughout any time-point during the course of AD, or at any stage of cognitive failure for that matter. Rather, it is usually a quantifiable up-regulation or down-regulation of an already existing biomarker in a specific anatomical region or biofluid compartment that has been the most consistently observed in the progressively degenerating brain. To cite a recent example, over 50 susceptibility genes and gene loci have been associated with late-onset AD and multiple models have been proposed [27], however, these associations are relatively rare and non-penetrant, occur in a few but not all AD cases, adding to the complexity and heterogeneity in the diagnosis of AD [15,56,58,63,64,69,70]. To further confound the establishment of definitive AD biomarkers, AD is commonly associated with more than one single neuropathology, in the case of AD usually with cerebrovascular and/or neurovascular involvement, and every one of these ancillary neurological disorders can carry their own set of complex and often overlapping disease biomarkers [10,63,64,69,70]. 

Especially over the last 10 years, the progressive and steadily increasing accumulation of molecular, genetic, epigenetic, neuroimaging, clinical, and geriatric data acquired from multiple AD cohorts has significantly increased our appreciation and understanding of the intrinsic variability and heterogeneity of AD biomarkers associated with the continuum of AD and other forms of progressive age-related neurodegenerative disease. The generation of massive datasets integrating multiple genetic-, epigenetic-, molecular-, and neuroimaging-derived biomarkers is enabling the application of clustering techniques and the identification and stratification of AD subtypes that may further categorize the multiple aspects of AD heterogeneity [10,11,12,13,14,15,16,17,18,67,68,69,70]. These approaches hold great potential: (**i**) for improving both the diagnosis and prognosis of AD; (**ii**) for projecting the clinical and neurological evolution of AD for planning suitable directions in therapeutic mediation; (**iii**) in providing multiple opportunities for the more directed analysis of AD heterogeneity in a data driven manner; (**iv**) in providing strategic guidelines for more decisive therapeutic intervention and the more efficacious clinical management of AD; and (**v**) for advancing *‘**precision medicine***’ not only for the individual AD patient, but also for other cases of inflammatory neurodegeneration and neurological disease.

### 3.2. Using Precision Medicine in the Diagnosis of AD

Multiple analytical molecular-genetic approaches, advances in geriatric psychiatry and clinical evaluation, advancements in neuro-radiological labelling techniques and neuroimaging technologies, integrated diagnostic and predictive strategies and methodological improvements, discoveries of the comprehensive pathophysiological profiles of complex multi-factorial neurodegenerative diseases: (**i**) are presently well within the capabilities and scope of contemporary clinical, medical, and diagnostic neurology; (**ii**) are currently yielding increasingly large volumes of biomarker data for both individual AD patients, large populations of AD cases and age- and gender-matched controls; and (**iii**) are providing a data-driven basis for the paradigm shift of using the ‘***precision medicine’*** approach in AD prevention, diagnostics, prognostics, and therapeutics [10,13,14,16,22,27,64]. Less common clinical presentations of AD are becoming increasingly recognized, adding to the increasing volume of AD biomarker data [10,14,17,19,64]. Since one of the pillars of ‘***precision medicine’*** is supported by biomarker-derived medical data, further improvements in the acquisition, integration, interpretation, and bioinformatics aspects of clinical data and the coordination and analysis of clinical, laboratory, molecular-genetic, and neuroimaging data, geriatric and psychological information and related healthcare resources should obtain significantly increased accuracy in the diagnostic synopsis for the “***prospective AD patient***”. The significant heterogeneity of the AD condition: (**i**) will certainly benefit from an equally wide variety of AD biomarker-derived ‘***precision medicine***’-oriented treatment approaches and/or data-driven pharmacological strategies; and (**ii**) whose biomarker-based therapeutic design will greatly improve the current situation regarding the healthcare, more effective and successful treatment, and the development of disease-modifying drugs for AD patients at any stage of the disease [10,22,65,68,71]. 

## 4. Summary

The ongoing search for valid biomarkers for AD is being carried out globally in at least a dozen major geriatric, bioinformatic, neurobiological, neuro-genetic and neurological bioscience arenas: (**i**) those involving the age, gender, and geriatrics of the ***‘prospective**AD patient’***; (**ii**) in the genetics and epigenetics of the AD patient including messenger RNA (mRNA) and microRNA (miRNA) signaling patterns, complexity and genomic methylation research**;** (**iii**) in multiple biofluids from AD patients including the blood (plasma/serum) of the systemic circulation, the glymphatic system, the cerebrospinal fluid (CSF) and/or urine; (**iv**); through the detailed analysis of molecular cargos from both biofluids and tissue-compartmentalized exosomes and extracellular microvesicles (EXs and EMVs); (**v**) throughout the peripheral nervous system (PNS; typically using skin biopsies); (**vi**) via clinically-based geriatric, psychiatric, and neurological assessment and testing; (**vii**) via advances in neuro-radiological labeling techniques and neuroimaging technologies including CAT, PET, PET-SN, MRI, fMRI; UHF-MRI, DOT, MEG, SPECT, cranial ultrasound, functional ultrasound (fUS) imaging, and immunohistochemistry involving confocal laser scanning microscopy and other advanced microscopic and neuroimaging techniques; (**viii**) from the quantitation and characterization of the load of microbial and microbial-derived components in the AD-affected brain; (**ix**) via the identification, quantitation, and characterization of AD-specific lesions including amyloid peptide-enriched SPs and NFTs; (**x**) after post-mortem examination and biopsies of AD cases, again matched up against those same biomarkers in age- and gender-matched neurologically normal controls to corroborate the prospective diagnosis of AD; (**xi**) via the comprehensive analysis of the potential contribution of overlapping progressive, age-related neurological disorders to AD-type change; and lastly (**xii**), through the assessment of the socioeconomic, environmental, and lifestyle factors of the ***‘prospective**AD patient’*** (Table 1). The recent application of highly integrated data sets of these multiple AD biomarkers and analytical techniques on large cohorts of AD patients and involving systems-biology and **‘*precision medicine’*** may well serve to unravel many of the more intricate aspects of AD heterogeneity and expand and build on current therapeutic strategies to more effectively address both the diagnosis and clinical management of this devastating neurological disease.

## Figures and Tables

**Table 1 jpm-10-00138-t001:** **Multiple interrelated factors contribute to AD.** The considerable heterogeneity of Alzheimer’s disease (AD) appears to be mediated in part by a highly interconnected network of biological factors, and each of these can be used as diagnostic biomarkers which appear to each have a variable potential to contribute to AD-type change. There is abundant evidence that all 23 of these biomarkers and/or factors (listed alphabetically) contribute to AD initiation, onset, or propagation, and there may be other important biological factors and other elements that may contribute to this complex neurological disease that we have not yet recognized or even considered. Data derived from each of these multiple biomarkers and factors combined is amenable to systems and network analysis, information integration, and the application of ***precision medicine*** that should ultimately yield a more accurate diagnosis of AD at any stage of the disease (see text for references; specific references to the biomarkers listed in Table 1 can be found in [15,16,17,18,19,20,21,22,23,24,25,26,27,28,29]. Abbreviations: BACE = β-amyloid cleavage enzyme; CRP = C-reactive protein (a blood-serum-based inflammatory biomarker); lncRNA = long non-coding RNA; PSI, PS2 = presenilin 1, presenilin 2; rRNA = ribosomal RNA; sncRNA = sall non-coding RNA.

age and age-related effects;
amyloid (Aβ40 and Aβ42 peptides);
compartmentalization of biomarkers [brain tissue, extracellular fluid (ECF), CSF, blood serum, urine];
cytokine storm (cytokines and chemokines);
environment and environmental effects;
epigenetics (methylation, mRNA and miRNA editing);
exosomes and extracellular micro-vesicles (EXs and EMVs);
gender and gender-related effects;
genetics (mutations in BACE, PS1, PS2, etc.,);
gastrointestinal (GI) tract microbiome;
innate immunity;
Neuro-inflammatory markers (CRP);
inter-current illness (cardiovascular disease);
lifestyle (diet, smoking);
messenger RNA (mRNA);
microbial contribution (viral, bacterial, fungal, other);
microbiome (oral, other);
microRNA (miRNA);
miRNA-mRNA linking patterns;
misdiagnosis;
oral microbiome and dental hygiene;
other RNA (sncRNA, lncRNA);
overlapping neurological disorders: [Downs syndrome (Trisomy 21), frontotemporal dementia (FTD), multi-infarct dementia (MID), neuro-vascular disease, prion disease (PrD), etc.,].

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
