# Peer review of "Biomarkers for Alzheimer’s Disease (AD) and the Application of Precision Medicine"

_jpm, 2020, doi:10.3390/jpm10030138_

Round 1

Reviewer 1 Report

Overall there are small grammatical errors throughout that need to be addressed as well as a significant amount of run on sentences that don't allow for clarity of concepts. (eg first sentence within the overview, first sentence of first and second paparagraph in challenges in the validation of AD biomarkers section).

This review could benefit from representative figures that allow the reader to visualize how diagnostic biomarkers are used (eg cartoon figure of EX and EMV AD biomarkers)

Authors should also make sure to briefly describe various concepts and technologies as well as expand and make clear various results of studies cited here (eg NF-kB was found to be unregulated in AD brain -but in human studies? animal? through CSF detection?)

Perhaps to better strengthen the review and ad novelty the authors may want to suggest current clinical therapies available along with listed biomarkers to better exemplify the strength of precision medicine. 

Author Response

Comments and Suggestions for Authors

Tuesday 8 September 2020

Firstly we would like to thank Reviewer #1 for their valuable time and expertise in the review of our invited Commentary article for potential publication in the"Journal of Personalized Medicine" (JPM).

Please find below the ‘RESPONSE’ to each of the Reviewers ‘COMMENTs’. We feel that the changes, additions and clarifications made including the addition of a Table 1 and 5 new 2020 references has greatly strengthened our contribution to the Special Issue entitled "Novel Biomarkers in Alzheimer's disease", in collaboration with "Journal of Personalized Medicine" (JPM).

Sincerely

Walter J. Lukiw BS, MS, PhD, Louisiana State University Health Sciences Center

===================================================================================

Comments of Reviewer #1

[1] COMMENT - Overall there are small grammatical errors throughout that need to be addressed as well as a significant amount of run on sentences that don't allow for clarity of concepts. (eg first sentence within the overview, first sentence of first and second paparagraph in challenges in the validation of AD biomarkers section).

[1] RESPONSE – This ‘Commentary paper’ has been gone over carefully by a medical-scientific editor (with degrees in the English language) to remove run on sentences and smooth out and clarify the message – thank you

[2] COMMENT - This review could benefit from representative figures that allow the reader to visualize how diagnostic biomarkers are used (eg cartoon figure of EX and EMV AD biomarkers)

[2] RESPONSE – A recent paper from our lab has just been accepted which is an extensive review of biomarkers in AD (with emphasis on miRNA biomarkers) and contains 3 detailed Figures:

Zhao, Y., Jaber V., Alexandrov, PN, Vergallo A., Lista S., Hampel H., Lukiw W.J. microRNA-based biomarkers and Alzheimer’s disease (AD) Frontiers in Neuroscience – Neurodegeneration - Special Research Topic ‘Deciphering the biomarkers of Alzheimer’s disease’ in press (2020).

To avoid redundancy and repetition and to allay Reviewer 1’s concerns in the current paper this item has been provided in the form of a Table 1 of biomarkers and related factors which appear to contribute to the onset or propagation of AD. There are now listed 14 references in the legend to Table 1 which discuss the biomarkers in Table 1 in some detail;

Further, in the revised manuscript text we direct the reader’s attention to another recent paper from our lab which illustrates a cartoon-type figure depicting EX and EMV as potential AD biomarkers; please see:

Lukiw WJ, Pogue AI. Vesicular transport of encapsulated microRNA between glial and neuronal cells. Int J Mol Sci. 2020;21(14):5078. Published 2020 Jul 18. doi:10.3390/ijms21145078

Again, to avoid redundancy and repetition this 2020 paper was referenced in the text of the current Commentary paper.

[3] COMMENT - Authors should also make sure to briefly describe various concepts and technologies as well as expand and make clear various results of studies cited here (eg NF-kB was found to be unregulated in AD brain -but in human studies? animal? through CSF detection?)

[3] RESPONSE – this item has been addressed in the revised manuscript text; NF-KB has been found to be significantly u-regulated in all AD human and animal studies yet undertaken;

[4] COMMENT Perhaps to better strengthen the review and ad novelty the authors may want to suggest current clinical therapies available along with listed biomarkers to better exemplify the strength of precision medicine. 

[4] RESPONSE

The concept of ‘Precision medicine’ in still in the earliest stages of development and its full potential is yet to be realized and implemented;

It was one of the goals of this Commentary paper to review the current list of available AD biomarkers which are, as it states in the revised Commentary text,

‘… one of the pillars of ‘precision medicine’ is supported by biomarker-derived medical data …’

Tuesday 8 September 2020

Lastly we would like to again thank Reviewer #1 for their valuable time, expertise and insight in the review of our manuscript. We sincerely believe that the additions-corrections-clarifications that we have made and the 5 additional references added to our current paper has considerably strengthened this invited contribution to Special Issue entitled "Novel Biomarkers in Alzheimer's disease", in collaboration with "Journal of Personalized Medicine" (JPM).

Sincerely

Walter J. Lukiw BS, MS, PhD, Professor of Neurology, Neuroscience and Ophthalmology, Bollinger Professor of Alzheimer’s disease (AD), LSU Neuroscience Center, Louisiana State University Health Sciences Center, New Orleans LA 70112 USA TEL 504-599-0842 EMAIL [email protected]

Reviewer 2 Report

This is a commentary on biomarkers for Alzheimer’s disease and application of the precision medicine. This article talks about the recent advances in biomarker research, challenges in the validation of AD biomarkers, and the need for precision medicine AD. The authors present the latest evidence to make their point.

  • The ideas in this article are not presented in a cohesive manner. Various sections of the paper, the subsections especially, begin without any introduction to the topic, leaving the reader to play catch-up. For example, the section on imaging technologies (beginning line 250) could begin with a sentence or two that indicate what imaging signs we typically look for/see in a patient with AD. While the general flow of the paper could be followed, there was little continuity from one subsection to the next
  • The abstract needs to be worked on. The language and terminology used are confusing and for e.g. “Noninvasive patient biofluids” and “AD is typically diagnosed by integrated knowledge? etc.”
  • I would suggest presenting challenges in the validation of the AD biomarker section after the authors talk about the sections on advances in biomarkers. This could be placed just before the section on precisions medicine in the diagnosis of AD.
  • Advanced, novel, and emerging diagnostic biomarker for AD: What tissue is used to source these biomarkers? Is this postmortem study?
  • AD biomarker and Post-mortem neuropathological examination of the AD brain: the lines from 2709-283 are redundant as they are repetitious.
  • My main critique was in the last section before the summary(starting line 312). This section focused on Precision Medicine in diagnosing AD. It feels as though the section spoke about the concept of precision medicine at length but was lacking in more specific examples of what would be included in the precision medicine paradigm shift. 

Author Response

Comments of Reviewer #2

Tuesday 8 September 2020

Firstly we would like to thank Reviewer #2 kindly and sincerely for their valuable time and expertise in the review of our Commentary article. Please find below the ‘RESPONSE’ to each of the Reviewers ‘COMMENT’ below.

We feel that the changes, additions and clarifications made, including the addition of a Table 1 and 5 very recent (2020) references has greatly strengthened our contribution to the Special Issue entitled "Novel Biomarkers in Alzheimer's disease", in collaboration with "Journal of Personalized Medicine" (JPM).

===================================================================================

This is a commentary on biomarkers for Alzheimer’s disease and application of the precision medicine. This article talks about the recent advances in biomarker research, challenges in the validation of AD biomarkers, and the need for precision medicine AD. The authors present the latest evidence to make their point.

[1] COMMENT

The ideas in this article are not presented in a cohesive manner. Various sections of the paper, the subsections especially, begin without any introduction to the topic, leaving the reader to play catch-up. For example, the section on imaging technologies (beginning line 250) could begin with a sentence or two that indicate what imaging signs we typically look for/see in a patient with AD. While the general flow of the paper could be followed, there was little continuity from one subsection to the next

[1] RESPONSE

We had assumed that anyone who chooses to read this Commentary article already has a rudimentary background in AD and a general idea of the imaging diagnostics used for this common neurological disease.

To allay Reviewer #2’s concerns we have taken their advice and added some relevant information (beginning at line 250); in addition five recent references have been added that review current imaging and AD biomarker technologies and their purpose, rationale for implementing them and their merits; (see Kamagata et al., 2020; Lo Buono et al., 2020; Lombardi et al., 2020; Vernooij and van Buchem 2020 and Zhao et al., 2020).

[2] COMMENT The abstract needs to be worked on. The language and terminology used are confusing and for e.g. “Noninvasive patient biofluids” and “AD is typically diagnosed by integrated knowledge? etc.”

[2] RESPONSE This has been corrected in the revised manuscript text. The entire ‘Commentary paper’ has been gone over carefully by a medical-scientific editor (with degrees in the English language and 30 year knowledge of neuroscience) to smooth out and clarify the message. In addition the entire manuscript has been proof read by 2 independent neuroscientist researchers knowledgeable in AD and AD biomarker research.

[3] COMMENT - I would suggest presenting ‘Challenges in the validation of the AD biomarker’ section after the authors talk about the sections on advances in biomarkers. This could be placed just before the section on precisions medicine in the diagnosis of AD.

[3] RESPONSE – As suggested by Reviewer #2 this has been corrected in the restructured and revised manuscript text.

[4] COMMENT - Advanced, novel, and emerging diagnostic biomarker for AD: What tissue is used to source these biomarkers? Is this postmortem study?

[4] RESPONSE

This has been corrected and clarified in the revised manuscript text – these studies involved CSF, blood serum and post-mortem tissues.

[5] COMMENT AD biomarker and Post-mortem neuropathological examination of the AD brain: the lines from 2709-283 are redundant as they are repetitious.

[5] RESPONSE

This has been also been corrected and clarified in the revised manuscript text. As suggested by Reviewer #2 relevant information has been added to streamline the essential message of this invited Commentary paper.

[6] COMMENT My main critique was in the last section before the summary(starting line 312). This section focused on Precision Medicine in diagnosing AD. It feels as though the section spoke about the concept of precision medicine at length but was lacking in more specific examples of what would be included in the precision medicine paradigm shift. 

[6] RESPONSE

We understand that the goal of publishing a ‘Commentary’ article is to advance the research field by providing a forum for varying perspectives on a certain topic under consideration in the journal by the authors. Typically the authors of a 'Commentary article' have in-depth knowledge of the topic and are eager to present a new and/or unique viewpoint on existing problems, fundamental concepts, prevalent notions and aim to discuss the implications of a newly implemented innovation and/or advancement in technologies and methodologies. A commentary may also draw attention to current advances and speculate on future directions of a certain topic, and may include original data as well as state a personal opinion.

This revised paper represents a commentary on ‘Biomarkers for Alzheimer's disease (AD) and the application of Precision Medicinebased on all of the authors’ long standing experiences in AD and AD biomarker research from multiple perspectives. The experience of each of the authors in AD research is in the time frame of 15-55 years per person. The concept of ‘precision medicine’ involves a relatively new and evolving network approach, and due to the extreme heterogeneity of AD biomarkers may be our best paradigm yet for the most accurate and definitive prediction, diagnosis and prognosis of AD. It should again be mentioned that ‘precision medicine’ is based on the acquisition, integration and networking of as much AD biomarker data as possible to generate a more accurate diagnosis of this insidious and lethal brain disorder, and this was the basic purpose for writing this Commentary article.  

===================================================================================

Tuesday 8 September 2020

Lastly we would again like to sincerely thank Reviewer #2 for their valuable time and expertise in the review of our Commentary article. We genuinely feel that the changes, additions and clarifications made, including the addition of a Table 1 and five highly supportive 2020 references, has greatly strengthened our contribution to the Special Issue entitled "Novel Biomarkers in Alzheimer's disease", in collaboration with "Journal of Personalized Medicine" (JPM).

Sincerely

Walter J. Lukiw BS, MS, PhD (corresponding author), Louisiana State University Health Sciences Center, New Orleans LA 70112 USA, TEL 504-599-0842, EMAIL [email protected]

===================================================================================

END OF COMMENTS - RESPONSES

Round 2

Reviewer 2 Report

Thank you for incorporating the changes and revising the manuscript.  The flow and the narrative of the article have improved a lot.  

I am not sure if table 1 adds value to the article. However, the title of the table is missing and check for the typos in the legend.